# Relationship between Chronic Rhinosinusitis and the Incidence of Head and Neck Cancer: A National Population-Based Study

**DOI:** 10.3390/jcm11185316

**Published:** 2022-09-09

**Authors:** Kyung-Do Han, Sang-Hyun Park, Sumin Son, Seung-Ho Kim, Ikhee Kim, Jong-Yeup Kim, Seung-Min In, Yeon-Soo Kim, Ki-Il Lee

**Affiliations:** 1Department of Statistics and Actuarial Science, Soongsil University, Seoul 03080, Korea; 2Department of Medical Statistics, College of Medicine, Catholic University of Korea, Seoul 03080, Korea; 3Department of Otorhinolaryngology-Head and Neck Surgery, Konyang University College of Medicine, Daejeon 35365, Korea; 4Department of Biomedical Informatics, Konyang University College of Medicine, Daejeon 35365, Korea; 5Myunggok Medical Research Institute, Konyang University College of Medicine, Daejeon 35365, Korea

**Keywords:** sinusitis, head and neck cancer, prognosis, cohort studies, Korea

## Abstract

We analyzed the relationship between chronic rhinosinusitis (CRS) and the incidence of head and neck cancers (HNCs) in a Korean adult population. This retrospective cohort study included data from the Korean National Health Insurance Service database. Adjustments were made to minimize risk variables for sex, age, diabetes, hypertension, dyslipidemia, and rhinitis between the two groups. The primary endpoint was newly diagnosed HNC between January 2009 and December 2018. Among 1,337,120 subjects in the Korean National Health Insurance Service database, data from 324,774 diagnosed with CRS (CRS group) and 649,548 control subjects (control group) were selected. Patients with CRS exhibited a statistically significant greater risk for nasal cavity/paranasal sinus cancer, hypopharynx/larynx cancer, and thyroid cancer compared with the control group. In the CRS group, the adjusted hazard ratios for nasal cavity/paranasal sinus cancer were 1.809 (95% confidence interval (CI) 1.085–3.016), 1.343 (95% CI 1.031–1.748) for hypopharynx and larynx cancer, and 1.116 (95% CI 1.063–1.173) for thyroid cancer. CRS was associated with a higher incidence of HNCs. Therefore, physicians should carefully consider the possibility of HNC progression and implement therapeutic strategies to minimize the impact of these diseases.

## 1. Introduction

Chronic rhinosinusitis (CRS) is one of the most prevalent upper respiratory diseases encountered in otorhinolaryngology clinics [1]. In general, CRS affects quality of life (QoL) and can be associated with other systemic diseases [2]. Associations between CRS and other systemic diseases, such as allergic [3], respiratory [4], gastrointestinal [5], and even neurodegenerative [6] diseases, have been previously reported. Considering their effects on the overall QoL, it can be hypothesized that CRS is also associated with head and neck cancers (HNCs). Recent studies have demonstrated an association between several diseases and HNC [7,8,9]. In a population-based study, Seo et al. [7] reported that hypertension was associated with the risk of oral, laryngeal, and esophageal cancers. In addition, Kim et al. [8] demonstrated that metabolic syndrome was significantly associated with the development of laryngeal cancer. In contrast, Jiang et al. [9] reported that metabolic syndrome did not increase the risk for HNC. This inconsistent outcome may be due to methodological discrepancies, such as different subject groups or incomplete control of confounding factors. As such, there is a lack of consensus regarding the effects of underlying diseases on HNCs.

Few studies have examined the relationship between CRS and HNC. In a case-cohort study involving an elderly population in the United States, CRS was associated with nasopharyngeal cancer (NPC) and nasal cavity and paranasal sinus cancer (NCPSC) [10]. Similarly, Kim et al. [11] reported that the relative risk for NPC and NCPSC was significantly greater in older patients with nasal polyps than in control subjects. However, these studies also had methodological weaknesses, such as limited subjects (elderly patients) and a specified phenotype (nasal polyps). Thus, despite previous research, the risk for HNC in those with CRS has not yet been determined. Therefore, the aforementioned hypothesis linking CRS and HNC needs to be clarified in a large population with adjustment for risk factors to overcome methodological limitations.

Data from the Korean National Health Insurance Service (NHIS) database have recently become available for medical analyses. This database houses large-scale information, covering approximately 50 million individuals in South Korea. Accordingly, our study aimed to identify whether patients with CRS are at higher risk for HNC by analyzing large-cohort datasets.

## 2. Materials and Methods

### 2.1. Data Source

This retrospective cohort study was performed using data from the NHIS claims database. The NHIS is a governmental organization that manages the national insurance system in Korea, which covers >95% of the entire Korean population [12,13,14]. The database houses information such as basic demographics, diagnosis, prescription or procedure, medical costs, and hospital or department. In particular, diagnoses are based on the International Classification of Diseases, 10th Revision (ICD-10) codes and all inpatient and outpatient claim information can be analyzed. The Korean population has a resident registration number and most individuals are supported by the Korean insurance system. Information from the NHIS database covers most Korean citizens and remains accurate because every hospital is required to use the resident registration number before practice. Originally, the NHIS platform was created for insurance claims and nationwide statistics of medical management. However, retrospective cohort studies of large populations using the NHIS database have been performed in South Korea since data from the NHIS have recently become available for medical research.

### 2.2. Ethics Approval

The Institutional Review Board (IRB) of Konyang University Hospital (Daejeon, Korea) approved this study (2020-05-006). Due to the retrospective design of the study and use of anonymized subject data, the need for written informed consent was waived by the IRB.

### 2.3. Study Population and Setting

Data of subjects diagnosed with CRS between January 2005 and December 2018 were collected from the NHIS database. Patients with CRS were defined as those who visited the outpatient clinic more than two times 3 months after diagnosis with an ICD (International classification of diseases) code of J32 and underwent nasal endoscopy (E7530, E7540, E7550 or E7560) (Table 1).

Baseline characteristics, including age, sex, urban residency, and income level (lowest quintile) were analyzed. In addition, comorbidities, such as diabetes (ICD-10 code E11–14 + ≥2 outpatient visits or ≥1 admission within 1 year), hypertension (ICD-10 code I10–I13 or I15 + ≥2 outpatient visits or ≥1 admission within 1 year), dyslipidemia (ICD-10 code E78 + ≥2 outpatient visits or ≥1 admission within 1 year), rhinitis (ICD-10 code J30 + ≥3 outpatient visits within 1 year), and nasal septal deviation (ICD-10 code J34.2 + nasal endoscopy, and nasal polyps (ICD-10 code J33 + nasal endoscopy) were analyzed to control for risk factors (Table 1). These diagnostic definitions were modified as previously reported [2,15,16,17,18]. Multivariate analysis was subsequently performed for age, sex, income level, systemic comorbidities, and rhinitis. 

Four models were used in this study: the first was non-adjusted; the second was adjusted only for age and sex; the third was completely adjusted for age, sex, income level, and basic comorbidities (i.e., hypertension, diabetes, and dyslipidemia); and, finally, the fourth included adjustment for rhinitis and CRS.

### 2.4. Statistical Analysis

Categorical variables are expressed as frequency and percentage and numerical data are expressed as mean ± standard deviation (SD). The crude incidence rates for HNC are expressed as the number of events per 1000 person-years. Kaplan–Meier plots reflect the cumulative incidence of each HNC between the CRS and control groups. The Cox proportional hazard model was performed for the adjustment to determine the hazard ratio (HR) for CRS regarding the incidence of each HNC. The HRs were analyzed using 95% confidence intervals (CI).

All statistical analyses were performed using SAS version 9.4 (SAS Institute, Inc., Cary, NC, USA) and R version 4.0.3 (The R Foundation for Statistical Computing, Vienna, Austria). All analyses were performed using two-tailed 95% CIs and statistical significance was set at *p* < 0.05.

## 3. Results

### 3.1. Baseline Characteristics of the Subjects

Between 1 January 2009 and 31 December 2009, 1,337,120 individuals were newly diagnosed with CRS, of whom 324,774 were evaluated. As a control group, 649,548 participants were recruited from the entire national population of Korea (approximately 50 million individuals) (Figure 1). We enrolled a control group, which was age and sex matched with selected CRS subjects from the national population. To reflect the actual clinical scenario, we included twice as many individuals in the control group as in the CRS group (1:2 propensity matching). The participants’ demographic information is summarized in Table 2. Of the sample, 41.11% were male and the mean age was approximately 45.31 years at the time of diagnosis. Urban residency was revealed as significantly higher in the CRS group compared to the control group. There was also a statistically significant, though only slightly higher, proportion of patients with diabetes, hypertension, and dyslipidemia in the CRS group than in the control group. Meanwhile, patients with CRS had a higher frequency of rhinitis than those in the control group. Detailed results are summarized in Table 2.

### 3.2. Comparison of the Incidence of HNCs between the CRS and Control Groups

HNCs occurred more frequently in the CRS group than in the control group. The Kaplan–Meier plot revealed that the incidence of HNC was higher in the CRS group than in the control group over time (Figure 2) and this outcome was observed in all four models. In model 4, the HRs for NCPSC, hypopharyngeal/laryngeal cancer (HPLC), and thyroid cancer in the CRS group were 1.809 (95% CI 1.085–3.016), 1.343 (95% CI 1.031–1.748), and 1.116 (95% CI 1.063–1.173), respectively. Regarding oral/oropharyngeal, salivary gland, and NPC, HRs in the CRS group exhibited increased tendencies without statistical significance compared to the control group. The detailed results are summarized in Table 3.

### 3.3. Incidence of HNCs among Subjects with CRS According to Several Covariates

The incidence of HNCs varied according to several covariates. Interestingly, CRS patients with nasal polyps revealed higher HRs for NCPSC compared to those with CRS without nasal polyps (Table 4).

According to sex, HR values for NCPSC and HPLC were greater among men than in women. However, contrary to general expectations, there were no differences between the sexes in other HNCs. Regarding age, the HRs for oral/oropharynx, NPCSC, and HPLC were higher in subjects >50 years of age, as expected. In addition, the HR for NCPSC was higher in subjects living in urban areas. In contrast, the HRs for HNCs did not increase with comorbidities, such as diabetes, hypertension, and dyslipidemia. Interestingly, the HRs for HNCs were 1.813 (95% CI 1.07–3.071), 2.517 (95% CI 1.41–4.493), 3.446 (95% CI 1.486–7.992), and 1.799 (95% CI 1.242–2.606) for NCPSC without diabetes, HPLC with diabetes, NCPSC with hypertension, and HPLC with hypertension, respectively. Moreover, the HRs for oral/oropharyngeal cancer increased in patients without rhinitis, whereas the HRs for HPLC increased in patients with rhinitis. Regardless of systemic risk profile, the HRs for thyroid cancer were high. The detailed outcomes are summarized in Table 5.

## 4. Discussion

To the best of our knowledge, the present study was the first to evaluate the risk of developing HNCs among a large cohort of patients with CRS. This nationwide cohort study included 324,774 patients with incident CRS and 649,548 matched controls. Interestingly, over time, the incidence of HNCs among patients with CRS was found to increase at a faster rate than that in the control group. In particular, the risks for NPC, NPCSC, and HPLC were significantly greater than those in the control group, even after controlling for risk factors, such as chronic systemic diseases and rhinitis.

Several studies investigating the relationship between CRS and HNCs have been performed using population-based data. A previous study involving a sample of the elderly population in the United States reported an increased risk for HNC among patients with CRS (adjusted HR 1.37 (95% CI 1.27–1.48) for overall HNC, 3.71 (95% CI 2.75–5.02) for NPC, and 5.49 (95% CI 4.56–6.62) for NCPSC, respectively [10]. Kim et al. [11] demonstrated that the incidence rate ratio for subjects with nasal polyps compared to the control group was 7.00 (95% CI 5.28–9.25) for NCPSC and 1.78 (95% CI 1.28–2.42) for NPC. However, these previous studies were limited by insufficient controls and vague definition(s) of risk factors. More importantly, these studies focused only on a limited population, such as the elderly or patients with nasal polyps. However, we demonstrated that the risk of HNC in patients with CRS was higher than that in control subjects after adjusting for risk variables for individual HNCs.

Our investigation has several strengths compared with previous studies. First, we analyzed a customized dataset from the entire national population of Korea (approximately 50 million individuals) instead of using a representative national sample dataset of 1 million. Second, our analysis involved an adjusted model accounting for the risk profiles for HNC (e.g., diabetes, hypertension, and dyslipidemia) and rhinitis. Third, our results were analyzed based on incident cases using National Health Screening Program data, which reflect authentic clinical information.

The mechanism through which CRS contributes to HNC is multifactorial. First, CRS can continuously and directly stimulate the head and neck area owing to inherent symptoms of postnasal drip [19,20]. It can be inferred, therefore, that chronic postnasal drip caused by CRS could continuously stimulate the head and neck area, thus, increasing the risk for HNC. Second, histologically, CRS induces microscopic changes, including impaired mucociliary transport and accompanying mucosal inflammation, in the sinonasal mucosa [21,22]. This histological alteration may be associated with the development of HNC. Third, immunologically, inflammatory cytokines and cells induced by CRS can promote the development of HNCs. CRS continuously invokes chronic mucosal inflammation by releasing inflammatory cytokines in the peripheral airway mucosa [23]. The local inflammatory reaction and released cytokines may induce HNC [24,25]. Fourth, patients with CRS tend to engage in relatively unhealthy behaviors, including increased smoking frequency and alcohol consumption, as compared with matched controls [26]. Hao et al. [27] reported that patients with CRS revealed higher rates of comorbidities, such as hypertension, diabetes, and dyslipidemia, and an increased incidence of cancer-related disease compared to the control group. In the present study, patients with CRS also exhibited worse rates of systemic comorbidities compared to the control group. Systemic comorbidities could be associated with an increased incidence of HNC.

In a case-control cohort study, Xia et al. [28] reported that the risk of HNC was significantly higher in patients with CRS than those in the control group (adjusted odds ratio (OR): 1.53, 95% CI: 1.33–1.75). Interestingly, compared to CRS patients without surgery, they demonstrated that the risk of HNC was higher in CRS patients receiving surgery. In another nationwide study, Huang et al. [29] demonstrated that CRS was associated with the risk of developing NPC (adjusted OR: 2.23; 95% CI, 1.61–3.09), whereas no significant association among CRS and NPC was shown in patients followed up for more than 1 year (adjusted OR: 1.16; 95% CI, 0.76–1.78). Likewise, we found that there was no significant relationship between CRS and NPC in long-term follow-up periods, while there was a significant association of CRS with the incidence of NCPSC and thyroid cancer.

Our study also demonstrated the effects of several covariates, including sex, age, residency, diabetes, hypertension, dyslipidemia, rhinitis, and nasal polyps. There was a difference in HRs between the CRS and control groups according to age, sex, and comorbidities. Sex-based discrepancies have been reported in patients with upper airway obstruction, which could be due to anatomical, hormonal, and endocrinological differences [30]. Among the CRS patients in our study, male sex was independently correlated with elevated risks for NCPSC and HPLC. However, we did not find a statistically significant difference between the sexes for the other HNCs. Meanwhile, the risk for most HNCs was higher in older patients, as expected. In addition, we found that urban residency is correlated with increased risk for NCPSC. Chronic systemic symptoms and illnesses could be associated with HNC [31]. In the present study, patients with chronic systemic disease exhibited substantially higher HRs for HNCs (e.g., hypertension for NCPSC, diabetes for HPLC, and hypertension for HPLC) than those without chronic diseases. In addition, rhinitis and CRS commonly coexist, which was also the case in our study. There was no significant difference for HNC risks according to the existence of rhinitis. This outcome suggests that CRS increases the risk for HNC, ruling out the effect of rhinitis. Furthermore, we found that nasal polyps are significantly associated with elevated risk for NCPSC. It can be understood that phenotypic severity in CRS leads to sinonasal malignancy.

The present study, however, has some limitations. First, disease severity could not be evaluated. Most previous studies investigating CRS did not classify the severity of CRS, even in randomized controlled studies [32,33]. This is a common limitation in studies investigating CRS because most cases of CRS are diagnosed based on clinical judgment. Second, we did not consider cancer staging or recurrence. Advanced or recurrent HNC may lead to different outcomes than early or primary cases. This is essential because the clinical progression and prognosis of HNC vary. Third, we did not analyze patients with CRS according to the presence of nasal polyps. A previous study reported an association between nasal polyps and HNC using the NHIS database [11], from which we demonstrated the relationship between CRS and HNC, regardless of the presence of nasal polyps. Last, tobacco use, alcohol consumption, and human papilloma virus status could not be analyzed in the present study because these variables must be merged from different databases, such as annual health checkup. In future, a large prospective study using hospital data is required to overcome these limitations and validate our findings.

## 5. Conclusions

Analyzing data from the NHIS claim database revealed a higher incidence of HNCs, especially for NCPSC, HPLC, and thyroid cancers, among patients with CRS than in the control group. Clinically, these findings should be considered to better inform diagnostic and therapeutic decisions for patients with CRS.

## Figures and Tables

**Figure 1 jcm-11-05316-f001:**
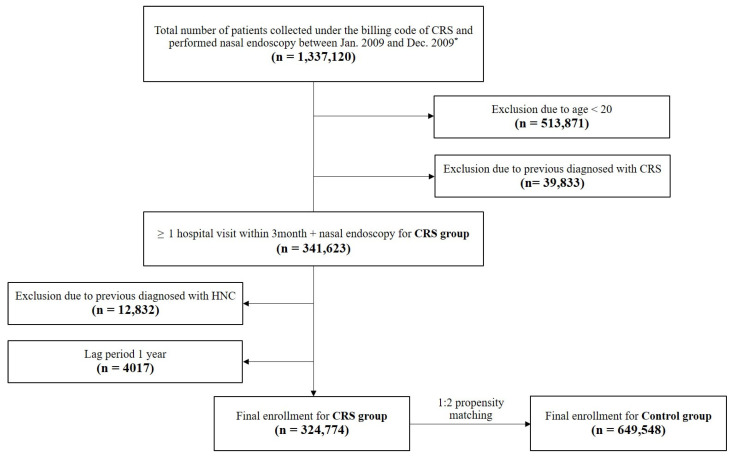
Schematic flow diagram illustrating subject selection for the present study. CRS: chronic rhinosinusitis. ***** At least one claim under ICD-10 codes J32 + nasal endoscopy (E7530, E7540, E7550 or E7560).

**Figure 2 jcm-11-05316-f002:**
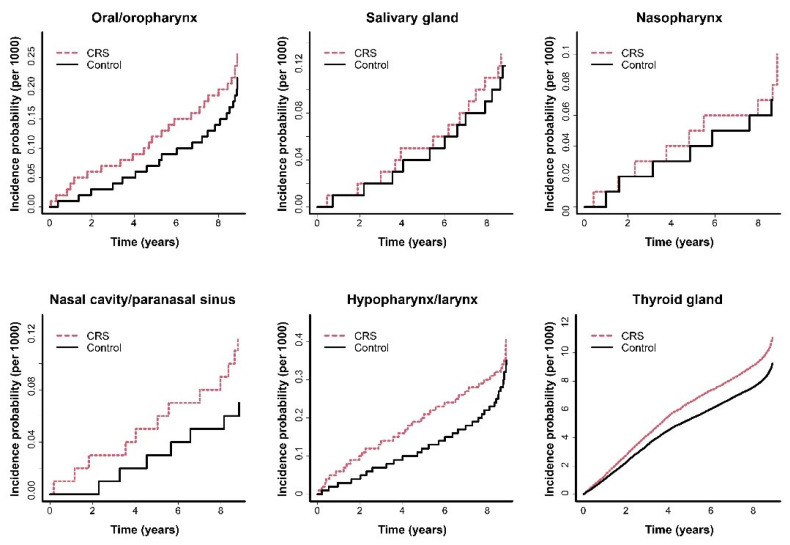
Kaplan–Meier curves for the incidence of head and neck cancer in patients with chronic rhinosinusitis (CRS). Head and neck cancers occurred more frequently in the CRS group compared with the control group.

**Table 1 jcm-11-05316-t001:** Working definition derived from the NHIS (National health insurance service) claims database.

Disease	Working Definition
Chronic rhinusinitis	J32 + ≥1 out-patient visit within 3 months + nasal endoscopy (E7530, E7540, E7550 or E7560)
Diabetes	At least one claim under ICD-10 E11-14 + ≥2 out-patient visits or ≥1 admission within 1 year
Hypertension	At least one claim under ICD-10 I10-13 or I15 + ≥2 out-patient visits or ≥1 admission within 1 year
Dyslipidemia	E78 + ≥2 out-patient visits or ≥1 admission within 1 year
Rhinitis	J30 + ≥3 out-patient visits within 1 year

ICD: international classification of diseases.

**Table 2 jcm-11-05316-t002:** Demographic characteristics of the CRS and control groups.

Variable	Control (n = 649,548)	CRS (n = 324,774)	*p* Value
Male	267,046 (41.11%)	133,523 (41.11%)	1.00
Mean age (year)	45.31 ± 15.35	45.31 ± 15.35	1.00
Income lowest quintile	138,945 (21.39%)	69,684 (21.46%)	0.46
Urban residency	306,502 (41.19%)	157,269 (48.42%)	<0.001
Diabetes	34,010 (5.24%)	19,565 (6.02%)	<0.001
Hypertension	107,391 (16.53%)	61,466 (18.93%)	<0.001
Dyslipidemia	50,800 (7.82%)	32,155 (9.9%)	<0.001
Rhinitis	83,857 (12.91%)	189,440 (58.33%)	<0.001

The data are expressed as counts (percentages) for categorical variables and as mean ± standard deviations for continuous variables.

**Table 3 jcm-11-05316-t003:** Crude and adjusted hazard ratios of overall CRS for HNC.

HNC	Groups	N	Events	Duration	Rate	Model 1	Model 2	Model 3	Mode 4
Oral/oropharynx	Control	649,548	124	5,416,889.91	0.0229	1	1	1	1
	CRS	324,774	71	2,650,612.95	0.0268	1.175 (0.878–1.574)	1.230 (0.919–1.648)	1.212 (0.904–1.624)	1.250 (0.8968–1.745)
Salivary gland	Control	649,548	73	5,416,889.91	0.0135	1	1	1	1
	CRS	324,774	38	2,650,612.95	0.0143	1.07 (0.723–1.584)	1.103 (0.745–1.633)	1.076 (0.726–1.594)	1.155 (0.739–1.804)
NPC	Control	649,548	44	5,416,889.91	0.0081	1	1	1	1
	CRS	324,774	29	2,650,612.95	0.0109	1.348 (0.843–2.154)	1.384 (0.866–2.213)	1.392 (0.870–2.226)	1.165 (0.672–2.019)
NCPSC	Control	649,548	41	5,416,889.91	0.0076	1	1	1	1
	CRS	324,774	38	2,650,612.95	0.0143	1.900 (1.222–2.954)	2.031 (1.305–3.160)	2.041 (1.311–3.179)	1.809 (1.085–3.016)
HPLC	Control	649,548	197	5,416,889.91	0.0364	1	1	1	1
	CRS	324,774	109	2,650,612.95	0.0411	1.135 (0.898–1.434)	1.259 (0.996–1.592)	1.247 (0.985–1.577)	1.343 (1.031–1.748)
Thyroid	Control	649,548	5651	5,416,889.91	1.0432	1	1	1	1
	CRS	324,774	3305	2,650,612.95	1.2469	1.195 (1.145–1.247)	1.186 (1.136–1.238)	1.186 (1.136–1.238)	1.116 (1.063–1.173)

Data are expressed as hazard ratio (95% confidence interval). Model 1 was non-adjusted; model 2 was adjusted by age and sex; model 3 was adjusted by age, sex, income level, diabetes, hypertension and dyslipidemia; and model 4: adjusted by age, sex, income level, diabetes, hypertension, dyslipidemia, and rhinitis. CRS, chronic rhinosinusitis; HNC, head and neck cancer; NPC, nasopharyngeal cancer; NCPSC, nasal cavity and paranasal sinus cancer; HPLC, hypopharyngeal/laryngeal cancer.

**Table 4 jcm-11-05316-t004:** Crude and adjusted hazard ratios of CRS for HNC according to nasal polyps.

HNC	Groups	N	Events	Duration	Rate	Model 1	Model 2	Model 3	Mode 4
Oral/oropharynx	CRSsNP	305,907	66	2,496,904.8	0.0264	1	1	1	1
	CRSwNP	18,867	5	153,708.15	0.03225	1.231 (0.496–3.056)	0.941 (0.378–2.344)	0.952 (0.382–2.371)	0.961 (0.379–2.441)
Salivary gland	CRSsNP	305,907	34	2,496,904.8	0.0136	1	1	1	1
	CRSwNP	18,867	4	153,708.15	0.026	1.908 (0.677–5.376)	1.597 (0.562–4.539)	1.595 (0.561–4.535)	1.608 (0.549–4.710)
NPC	CRSsNP	305,907	28	2,496,904.8	0.0112	1	1	1	1
	CRSwNP	18,867	1	153,708.15	0.0065	0.569 (0.077–4.185)	0.385 (0.052–2.836)	0.379 (0.051–2.795)	0.336 (0.045–2.526)
NCPSC	CRSsNP	305,907	30	2,496,904.8	0.012	1	1	1	1
	CRSwNP	18,867	8	153,708.15	0.052	4.340 (1.989–9.468)	3.321 (1.510–7.306)	3.332 (1.513–7.336)	3.140 (1.383–7.128)
HPLC	CRSsNP	305,907	103	2,496,904.8	0.0413	1	1	1	1
	CRSwNP	18,867	6	153,708.15	0.039	0.946 (0.416–2.154)	0.654 (0.2887–1.491)	0.655 (0.287–1.495)	0.750 (0.327–1.721)
Thyroid	CRSsNP	305,907	3140	2,496,904.8	1.2576	1	1	1	1
	CRSwNP	18,867	165	153,708.15	1.0735	0.848 (0.725–0.992)	1.143 (0.976–1.338)	1.144 (0.977–1.339)	1.112 (0.947–1.305)

CRS, chronic rhinosinusitis; CRSsNP, chronic rhinosinusitis without nasal polyps; CRSwNP, chronic rhinosinusitis with nasal polyps; HNC, head and neck cancer; NPC, nasopharyngeal cancer; NCPSC, nasal cavity and paranasal sinus cancer; HPLC, hypopharyngeal/laryngeal cancer.

**Table 5 jcm-11-05316-t005:** Hazard ratios (95% CIs) of various covariates regarding the incidence of HNC with CRS.

Variable		Oral/Oropharynx	Salivary Gland	NPC	NCPSC	HPLC	Thyroid
Sex	Male	1.224 (0.824–1.817)	1.515 (0.830–2.768)	1.540 (0.827–2.2866)	1.830 (1.014–3.302)	1.329 (1.011–1.747)	1.169 (1.045–1.307)
	Female	1.304 (0.768–2.215)	0.905 (0.499–1.642)	0.554 (0.197–1.561)	1.761 (0.753–4.123)	1.500 (0.668–3.366)	1.107 (1.051–1.167)
Age	<50	0.784 (0.429–1.435)	1.252 (0.658–2.383)	1.005 (0.445–2.273)	1.547 (0.571–4.194)	0.827 (0.344–1.988)	1.095 (1.033–1.161)
	≥50	1.475 (1.013–2.148)	1.095 (0.628–1.908)	1.267 (0.655–2.449)	1.858 (1.064–3.243)	1.371 (1.045–1.800)	1.161 (1.074–1.254)
Residency	Rural	1.035 (0.665–1.609)	1.217 (0.680–2.177)	1.505 (0.757–2.990)	1.117 (0.542–2.301)	1.473 (1.058–2.051)	1.158 (1.085–1.236)
	Urban	1.542 (0.983–2.418)	1.090 (0.594–2.001)	0.844 (0.387–1.839)	2.712 (1.392–5.285)	1.190 (0.814–1.742)	1.074 (1.006–1.147)
Diabetes	No	1.243 (0.876–1.762)	1.260 (0.795–1.998)	1.026 (0.578–1.820)	1.813 (1.070–3.071)	1.177 (0.882–1.572)	1.111 (1.056–1.168)
	Yes	1.314 (0.531–3.253)	0.462 (0.099–2.165)	6.620 (0.729–60.102)	1.769 (0.351–8.916)	2.517 (1.410–4.493)	1.243 (1.012–1.526)
Hypertension	No	1.098 (0.745–1.618)	1.006 (0.580–1.743)	1.186 (0.633–2.222)	1.303 (0.704–2.409)	1.064 (0.754–1.502)	1.102 (1.045–1.162)
	Yes	1.715 (0.975–3.015)	1.446 (0.743–2.817)	1.114 (0.440–2.2822)	3.446 (1.486–7.992)	1.799 (1.242–2.606)	1.188 (1.066–1.324)
Dyslipidemia	No	1.268 (0.877–1.833)	1.217 (0.749–1.978)	1.217 (0.689–2.148)	1.685 (0.977–2.907)	1.275 (0.957–1.698)	1.109 (1.054–1.167)
	Yes	1.190 (0.622–2.277)	0.935 (0.370–2.363)	0.728 (0.131–4.044)	2.776 (0.796–9.677)	1.713 (0.966–3.038)	1.192 (1.027–1.384)
Rhinitis	No	1.543 (1.056–2.254)	1.211 (0.722–2.034)	1.139 (0.563–2.302)	1.567 (0.827–2.969)	1.190 (0.857–1.654)	1.202 (1.131–1.279)
	Yes	0.793 (0.457–1.375)	1.027 (0.460–2.292)	1.207 (0.501–2.908)	2.476 (0.939–6.527)	1.757 (1.070–2.885)	0.992 (0.918–1.071)

CI, confidence interval; CRS, chronic rhinosinusitis; HNC, head and neck cancer.

## Data Availability

This study used the National Health Screening Cohort data (NHIS-2021-1-385) from the National Health Insurance Service. Data available on request from corresponding author.

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
