# Peer review of "Relationship between Chronic Rhinosinusitis and the Incidence of Head and Neck Cancer: A National Population-Based Study"

_jcm, 2022, doi:10.3390/jcm11185316_

Round 1

Reviewer 1 Report

The authors have presented a study about "The Relationship between Chronic Rhinosinusitis and the Incidence of Head and Neck Cancer: A National Population-Based Study".

The topic is interesting and the sample size considered is absolutely adequate.

The methods have been clearly described. 

I have some minor suggestions:

1) please provide higher quality figures for figure 2

2) it would be interesting to know if the Chronic Rhinosinusitis with or without polips have different risks of develping cancer in the H&N

3) tobacco is a strong risk factor and if the authors could find the way to obatin such information it would uhgely empower the results of the present report

Author Response

Point 1: please provide higher quality figures for figure 2.

Response 1: Thank you for the valuable comment. We enhanced the resolution from 300dpi to 600dpi for figure 2. In addition, we reformatted the boldness of numbers, texts and lines, and removed redandant lines to be more intuitive.

Point 2: it would be interesting to know if the Chronic Rhinosinusitis with or without polips have different risks of develping cancer in the H&N.

Response 2: We appreciate your nice comment. As the reviewer mentioned, risk assumption for H&N cancer according to phenotype (nasal polyp) would be interesting. We re-analyzed hazard ratios (95% CIs) subclassifying chronic rhinosinusitis according to nasal polyps (Table 4). We found that patients with nasal polyps revealed higher HRs for NCPSC compared to those with CRS without nasal polyps. In particular, it can be understood that phenotypic severity in CRS leads to sinonasal malignancy. We added this description in the Results and Discussion sections of the manuscript.

Point 3: tobacco is a strong risk factor and if the authors could find the way to obatin such information it would uhgely empower the results of the present report.

Response 3: We tried to add the information regarding smoking and also alcohol consumption status as the reviewer commented. Unfortunately, more detailed information regarding smoking and alcohol consumption is not possible in this retrospective cohort study usnig only diagnostic codes. It must be merged from different database such as annual health checkup. We can provide preliminary data regarding tobacco and alcohol consumption status using the annual health checkup database as below.

 In a future study, a retrospective study using the aforementioned database or a large prospective study using hospital data are required to overcome the limitation and validate our findings. We added this description in the Discussion section of the manuscript.

Control

CRS

pValue

N

147021

49007

Age

47.27±13.24

47.27±13.24

1

Sex(Male)

76974 (52.36)

25658 (52.36)

1

Income(Low)

31681 (21.55)

9405 (19.19)

<0.0001

Diabetis mellitus

10492 (7.14)

3904 (7.97)

<0.0001

Hypertension

27465 (18.68)

10201 (20.82)

<0.0001

Dyslipidemia

16246 (11.05)

6815 (13.91)

<0.0001

Nasal polyps

21576 (14.68)

34909 (71.23)

<0.0001

Nasal septal deviation

331 (0.23)

7712 (15.74)

<0.0001

Smoking

<0.0001

Non-smoker

89270 (60.72)

30727 (62.7)

Quitter

20980 (14.27)

8873 (18.11)

Current smoker

36771(25.01)

9407 (19.2)

Alcohol consumption

<0.0001

Non-drinker

76677(52.15)

26698 (54.48)

Moderate drinker

59021(40.14)

19233 (39.25)

Heavy alcoholics

11323(7.7)

3076 (6.28)

Reviewer 2 Report

The digits should be written with dots for readability, please. Instead of 138954 please 138,954

Why does the control group contain twice as many individuals as the CRS group? Isn't it better if the number is about equal?

Basically, it is difficult to create a statistical correlation with a few data from the population.

Many more variables would have to be included in the analysis instead of just the few variables in Table 2. 

The reported risk profiles for HNC, (diabetes, hypertension and dyslipidemia) are questionable. Virtually all known major risk factors are missing from the statistical evaluation : smokers/non-smokers, HPV infections, alcohol consumption.

The authors claim there was a higher proportion of diabetes, hypertension and dyslipidemia in CRS patients. However, according to Table 2, these are each very small percentages more 1.18% for dislipidemia, 2.4% hypertension, 0.78% diabetes). 

Although the authors claim in the discussion that they have a strong control group, virtually nothing is known about the control group except that it says "were recruited from the general population."

The groups also lack a lot of other information. For example, it is not clear how many people in the control group had other types of tumors. Or, perhaps more importantly, how many people live in urban areas and how many in rural areas (perhaps in urban areas the air pollution is much higher and therefore the risk of CRS much higher and at the same time the risk of tumors, This does not necessarily mean that CRS is related to HNC, but that the (more or less toxic) place of living is related.

Alternatively, which workplaces? How many people with CRS work in jobs where the respiratory tract is irritated? Compared to the control group? In the control group, perhaps more people work as farmers and not in toxic workplaces.

Again, the workplace could then be associated with increased HNC incidence and not primarily rhinusitis.

Table 2: Values are incorrectly reported.

The text says 41.11% men, the table says 21.39% men. If it is 41.11% men, then the number of men given in Table 2 is also incorrect.

Exactly the same number for Income lowest qunitile and number of men? Is this a coincidence or an error? 

Just like the "mean age" for Control and CRS group.

How can it be that the chronic rhinitis group has only 58.33% rhinitis? That should be close to 100%.

Author Response

Point 1: The digits should be written with dots for readability, please. Instead of 138954 please 138,954.

Response 1: Thank you for the valuable advice. We added dots for the digits in the manuscript as the reviewer suggested. We believe that adding dots enhance the readability.

Point 2: Why does the control group contain twice as many individuals as the CRS group? Isn't it better if the number is about equal?

Response 2: Thank you for important methodological question. We performed the retrospective cohort study using National Health Insurance Service database (NHIS). The NHIS claim data was originally made for the purpose of insurance coverage. Meahwhile, we collected the CRS group first and then enrolled the control group with matching age and sex. We tried to reflect the actual clinical clinical scenario. Therefore, we selected twice as many individuals as the CRS group (1:2 propensity matching) instead of 1:1 matching in order to enhance validation power. We revised this description in the Results and Discussion sections of the manuscript.  

Point 3: Basically, it is difficult to create a statistical correlation with a few data from the population.

Many more variables would have to be included in the analysis instead of just the few variables in Table 2.

Response 3: We added urban residency as you mentioned in Table 2. We also tried to add the variables such as nasal septal deviation and nasal polyps. However, diagnostic definitions of nasal polyps in this databse are too strict to collect enough samples. Meanwhile, urban residency revealed significantly higher in the CRS group compared to the control group. The HR for NCPSC was higher in subjects in urban area. We added the descriptions in the Results and Discussion sections of the manuscript.

Point 4: The reported risk profiles for HNC, (diabetes, hypertension and dyslipidemia) are questionable. Virtually all known major risk factors are missing from the statistical evaluation : smokers/non-smokers, HPV infections, alcohol consumption.

Response 4: The risk variables sich as diabetes, hypertension and dyslipidemia were analyzed in order to evaluate the role of systemic disorders. As far we know, these systemic variables have been analyzed in similar nationwide cohort studies.

(Please note Lee et al. Chronic rhinosinusitis increases the risk of hemorrhagic and ischemic stroke: A longitudinal follow-up study using a national sample cohort. PLOS ONE | https://doi.org/10.1371/journal.pone.0193886 March 1, 2018

and Cho et al. Uvulopalatopharyngoplasty Reduces the Incidence of Depression Caused by Obstructive Sleep Apnea Laryngoscope, 129:1005–1009, 2019)

Meanwhile, smoking and alcohol consumption status are known major risk factors as the reviewer mentioned. We tried to add the information regarding smoking and also alcohol consumption status as the reviewer commented. Unfortunately, more detailed information regarding smoking and alcohol consumption is not possible in this retrospective cohort study usnig only diagnostic codes. It must be merged from different database such as annual health checkup. We can provide preliminary data regarding tobacco and alcohol consumption status using the annual health checkup database as below.

 In a future study, a retrospective study using the aforementioned database or a large prospective study using hospital data are required to overcome the limitation and validate our findings. We added this description in the Discussion section of the manuscript.

Control

CRS

pValue

N

147021

49007

Age

47.27±13.24

47.27±13.24

1

Sex(Male)

76974 (52.36)

25658 (52.36)

1

Income(Low)

31681 (21.55)

9405 (19.19)

<0.0001

Diabetis mellitus

10492 (7.14)

3904 (7.97)

<0.0001

Hypertension

27465 (18.68)

10201 (20.82)

<0.0001

Dyslipidemia

16246 (11.05)

6815 (13.91)

<0.0001

Nasal polyps

21576 (14.68)

34909 (71.23)

<0.0001

Nasal septal deviation

331 (0.23)

7712 (15.74)

<0.0001

Smoking

<0.0001

Non-smoker

89270 (60.72)

30727 (62.7)

Quitter

20980 (14.27)

8873 (18.11)

Current smoker

36771(25.01)

9407 (19.2)

Alcohol consumption

<0.0001

Non-drinker

76677(52.15)

26698 (54.48)

Moderate drinker

59021(40.14)

19233 (39.25)

Heavy alcoholics

11323(7.7)

3076 (6.28)

Likewise, we tried to re-analyze the groups according to HPV infection status. However, we could not uncover the informative data. We understand that HPV infection is often coded incorrectly. We think that analyzing according HPV infection is methodologically inappropriate using our NHIS database.

Point 5: The authors claim there was a higher proportion of diabetes, hypertension and dyslipidemia in CRS patients. However, according to Table 2, these are each very small percentages more 1.18% for dislipidemia, 2.4% hypertension, 0.78% diabetes). 

Response 5: As the reviewer commented, there was only small percentage difference between two groups. On the other hand, pValue showed the statistical significance in each variable such as diabetes, hypertension, dyslipidemia and rhinitis. Therefore, we toned down the sentences in the manuscript accordingly as below.

There was a statistically significant, though only slight higher proportion of CRS patients with diabetes, hypertension, and dyslipidemia compared to the control group. Meanwhile, patients with CRS had a higher frequency of rhinitis than those in the control group.

Point 6: Although the authors claim in the discussion that they have a strong control group, virtually nothing is known about the control group except that it says "were recruited from the general population." 

Response 6: We added more detailed descriptions regarding control group selection in the Results section of the manuscript as seen below.

… were recruited from the entire national population of Korea (approximately 50 million individuals) (Figure 1). We enrolled a control group, which was age and sex-matched with selected CRS subjects, from the national population. To reflect the actual clinical scenario, we included twice as many individuals in the control group as in the CRS group (1:2 propensity matching).

Point 7: The groups also lack a lot of other information. For example, it is not clear how many people in the control group had other types of tumors. Or, perhaps more importantly, how many people live in urban areas and how many in rural areas (perhaps in urban areas the air pollution is much higher and therefore the risk of CRS much higher and at the same time the risk of tumors, This does not necessarily mean that CRS is related to HNC, but that the (more or less toxic) place of living is related.

Alternatively, which workplaces? How many people with CRS work in jobs where the respiratory tract is irritated? Compared to the control group? In the control group, perhaps more people work as farmers and not in toxic workplaces.

Again, the workplace could then be associated with increased HNC incidence and not primarily rhinusitis..

Response 7: Thank you for the nice comment. We agree with your opinion that environment or work place could serve as a critical confounder for HNC. We re-analyzed data and added information regarding urban residency accroding to the groups in Table 2. Urban residency reveled significantly higher in the CRS group compared to the control group. In addition, we found that urban residency is correlated with increased risk for NCPSC (Table 4). We also added these descriptions in the Results and Discussion sections in the manuscript.

 Meanwhile, we could not collect information regarding workplace using our NHIS database. It might be possible from other database such as National statistics. Analyzing HNC cancer risks according to workplace would be suitable topic for next study.

Point 8: Table 2: Values are incorrectly reported.

The text says 41.11% men, the table says 21.39% men. If it is 41.11% men, then the number of men given in Table 2 is also incorrect.

Exactly the same number for Income lowest qunitile and number of men? Is this a coincidence or an error? 

Just like the "mean age" for Control and CRS group.

Response 8: We are really sorry for this error. Number of men was wrong due to a technical error. We confirm that number of male are 267,046 (41.11%) and 133,523 (41.11%) in the control and the CRS group, respectively.

We also confirm that number of income lowest quintile is revised correctly (138,945 (21.39%) and 69,684 (21.46%), respectively).

Thank you for the great comment. We are really appreciate it.

Point 9: How can it be that the chronic rhinitis group has only 58.33% rhinitis? That should be close to 100%.

Response 9: As the reviewer mentioned, chronic rhinitis combined with chronic rhinosinusitis is very common clinically. However, this is retrospective cohort study using insurance claim codes from National health insurance system database. In order to improve the accuracy of the diagnostic code, the operational definition of subjects with chronic rhinosinusitis and chronic rhinitis in the present study was as follows: J32 + ≥ 1 out-patient visit within 3 months + nasal endoscopy (E7530, E7540, E7550 or E7560) and J30 + ≥ 3 out-patient visits within 1 year.

 Due to this strict diagnostic definition, the percentage of patients in the CRS group with chronic rhinitis as well was lower than one might expect.

 Although there is a discrepancy regarding the prevalence according to study design, other studies showed similar outcomes comparable with our study as seen below.

- Our study: the prevalence of rhinitis in general population, 12.91% and rhinitis combined with chronic rhinosinusitis, 58.33%, respectively.

- Bakheshaee et al.: 22.4% and 64%, respectively. (The Prevalence of Allergic Rhinitis in Patients with Chronic Rhinosinusitis. Iran J Otorhinolaryngol., Vol.26(4), Serial No.77, Oct 2014)

- Murat et al.: the prevalence of rhinitis combined with chronic rhinosinusitis, 71.2% (Rhinosinusitis among the patients with perennial or seasonal allergic rhinitis. Asian Pac J Allergy Immunol. 2003 Jun;21(2):75-8.)

Round 2

Reviewer 2 Report

The authors have made an effort to answer the given points. Partly they have convinced me. However, I remain unconvinced that the risk profiles, as the authors claim and on which they base the paper, for HNC are diabetes, hypertension, and dyslipidemia.(lines in manuscript 231-237). 

The main risk factors of HNC are, and this is known worldwide, smoking, alcohol abuse, HPV infection and betel nut chewing. 

The authors, referring to a literature review (Lee et al.), stated that CRS increases the risk of hemorrhagic and ischemic stroke. However, this is a rather weak example because the given paper examines CRS but not HNC. That may be true for CRS but not necessarily for HNC.  

The given paper uses risk profiles like hypertension, diabetes, dyslipidemia, ischemic heart disease, migraine, chronic kidney disease, depression, sleep disorder because these risks are also associated with strokes. But not with HNC.

And if it is true that CRS causes strokes, it does not mean that strokes cause CRS. 

Another literature site was cited by the authors as an example that uvulopalatopharyngoplasty lowers depression caused by sleep apnea. But the authors did not use depression, nor sleep apnea as a risk profile. Again, this may be true for palatopharyngeal surgery, but has little to do with the data presented.

Author Response

Point 1: The authors have made an effort to answer the given points. Partly they have convinced me. However, I remain unconvinced that the risk profiles, as the authors claim and on which they base the paper, for HNC are diabetes, hypertension, and dyslipidemia.(lines in manuscript 231-237). 

The main risk factors of HNC are, and this is known worldwide, smoking, alcohol abuse, HPV infection and betel nut chewing. 

Response 1: Thank you for your comment. We understand you are unconvinced regarding the classification as risk profiles for diabetes, hypertension, and dyslipidemia etc. for HNC in the present paper. We agree with your opinion that variables such as diabetes, hypertension, and dyslipidemia were not traditionally considered direct risk factors for HNC. The consensus well known major risk factors are smoking, alcohol abuse, HPV infection and betel nut chewing as you commented. Therefore, we toned down terminology from “risk factor” to just “systemic comorbidities” in the manuscript. In addition, we revised descriptions as below (previous lines in manuscript 231-237 area).:

Hao et al. reported that patients with CRS revealed higher rates of comorbidities, such as hypertension, diabetes, and dyslipidemia, and an increased incidence of cancer-related disease compared to the control group. In the present study, patients with CRS also exhibited worse rates of systemic comorbities compared to the control group. Systemic comorbidities could be associated with an increased incidence of HNC.

Point 2: The authors, referring to a literature review (Lee et al.), stated that CRS increases the risk of hemorrhagic and ischemic stroke. However, this is a rather weak example because the given paper examines CRS but not HNC. That may be true for CRS but not necessarily for HNC.  

The given paper uses risk profiles like hypertension, diabetes, dyslipidemia, ischemic heart disease, migraine, chronic kidney disease, depression, sleep disorder because these risks are also associated with strokes. But not with HNC.

And if it is true that CRS causes strokes, it does not mean that strokes cause CRS. 

Another literature site was cited by the authors as an example that uvulopalatopharyngoplasty lowers depression caused by sleep apnea. But the authors did not use depression, nor sleep apnea as a risk profile. Again, this may be true for palatopharyngeal surgery, but has little to do with the data presented.

Response 2: We also agree your comment that the two previously referenced papers could not be strong references for verifying the relationship between CRS and HNC risks. Hence, we cited more direct examples for CRS and HNC with similar methodology. We compared those studies with our outcomes in detail. We added one paragraph in the Discussion section of the manuscript for the comparison between previous studies and our study as below.:

In a case-control cohort study, Xia et al. reported that the risk of HNC was significantly higher in patients with CRS than those in the control group (adjusted odds ratio [OR]: 1.53, 95% CI: 1.33-1.75). Interestingly, compared to CRS patients without surgery, they demonstrated that the risk of HNC was higher in CRS patients receiving surgery. In another nationwide study, Huang et al. demonstrated that CRS was associated with the risk of developing NPC (adjusted OR: 2.23; 95% CI, 1.61-3.09) whereas no significant association among CRS and NPC was shown in patients followed up for more than 1 year (adjusted OR: 1.16; 95% CI, 0.76-1.78). Likewise, we found that there was no significant relationship between CRS and NPC in long-term follow-up periods, while there was a significant association of CRS with the incidence of NCPSC and thyroid cancer.